# Development and Evaluation of Psychometric Properties Regarding the Whole Person Health Scale for Employees of Hospital to Emphasize the Importance of Health Awareness of the Workers in the Hospital

**DOI:** 10.3390/healthcare9050610

**Published:** 2021-05-19

**Authors:** Chih-Ju Liu, Shih-Hsuan Pi, Chun-Kai Fang, Te-Yu Wu

**Affiliations:** 1School of Nursing, National Taipei University of Nursing and Health Sciences, Taipei 112303, Taiwan; thihju@mmh.org.tw; 2Department of Nursing, MacKay Memorial Hospital, Taipei 104217, Taiwan; 3Department of Medical Research, MacKay Memorial Hospital, Taipei 104217, Taiwan; chubby0224@hotmail.com; 4Hospice and Palliative Care Center and Department of Psychiatry, MacKay Memorial Hospital, Taipei 104217, Taiwan; 5Department of Thanatology and Health Counseling, National Taipei University of Nursing and Health Sciences, Taipei 112303, Taiwan; 6Department of Medicine, MacKay Medical College, New Taipei 26245, Taiwan; 7Department of Death Care Service, MacKay Junior College of Medicine, Nursing and Management, Taipei 11260, Taiwan; 8Department of Nursing, MacKay Medical College, New Taipei 26245, Taiwan

**Keywords:** whole person health, focus group, exploratory factor analysis, confirmatory factor analysis

## Abstract

(1) Background: Whole person health (WPH) is important among employees in hospitals. It will affect their performance and attitude toward patient care and organization. This project was designed to develop and assess the validity and reliability of utilizing the Whole Person Health Scale for Employees of a Hospital (WPHS-EH) to determine overall employee health. (2) Methods: A mixed-methods focus group and cross-sectional survey was adopted. Employees held six focus groups, with 62 employees from different departments in medical center in Taiwan. After analyzing the interview content, five experts tested its validity, and the 14-item WPHS-EH scale was analyzed. This was followed by an additional 900 participants questionnaire survey, response rate: 94.9%. Descriptive statistics, Cronbach’s alpha, exploratory factor analysis (EFA), and items analysis were used. Additionally, the scale was implemented to conducted confirmatory factor analysis (CFA) test for validity. (3) Results: Three dimensions were extracted from the questionnaires by EFA: “hospital circumstance and system”, “professional and interpersonal interaction” and “workload and harm”. The Cronbach’s alpha of the WPHS-EH scale was 0.82, while the three sub-dimensions were all significantly correlated with total scores. CFA confirmed the scale construct validity, with a good model fit. (4) Conclusions: The WPHS-EH is a reliable measurement tool to assess the effects of hospitals’ Whole Person Health among employees. The intent of the WPHS-EH was to provide a reliable scale to analyze the work environment for hospital staff and useful information to healthcare administrators interested in improving the staff’s whole person health.

## 1. Introduction

“Health” is an intricate balance of well-being comprising the following six dimensions: physical, psychological, emotional, social, spiritual, and group. These six categories can each be further divided to incorporate the following: the soundness of physical functions; clear and organized thoughts; the ability to deal with stress, depression, and anxiety; the ability to create and maintain relationships with others; the ability to achieve inner peace; and living in a healthy environment, respectively [1,2]. Health is based on the interplay among an individual’s physical, psychological, social, and environmental contexts and each dimension has its own importance and degree of influence. Whole person health, then, incorporates the following characteristics: feeling healthy and happy, getting along well with family and friends, working effectively, enjoying entertainment, maintaining self-empowerment, planning for the future, and keeping an optimistic attitude [3,4] Mental health, on the other hand, integrates aspects of realizing one’s happiness and potential, coping with normal stress in life, working effectively, and contributing to the community. It is broadly defined as a continuum, ranging from a negative state, such as depression or burnout, to a positive state, such as the presence of happiness [5]. In other words, whole person health is a comprehensive self-concept of a person’s overall health.

When employees are overworked, work pressure increases, leading to not only a decline in work efficiency, but also physical and mental exhaustion [6]. In Taiwan, and even throughout East Asia and Southeast Asia, the staff of medical institutions are always too busy. After the global outbreak of COVID-19, the health of medical staff needs to be taken seriously. As long as such pressure persists, employees may suffer the following consequences: alterations in physical and mental health, declining productivity, poor interpersonal relationships and morale, increased conflicts, reduction in group cohesion, higher turnover rate and accidents, and other behavioral changes that will indirectly affect team dynamics [6,7,8,9,10].

Medical institutions have to pay more attention to the importance of physical and mental health of medical staff because it is not only a simple self-care issue, but also affects the quality of patient care. Previous studies on medical and health working scenarios focused heavily on their negative impacts such as high work pressure, emotional exhaustion, fatigue, and low job satisfaction among hospital employees, and rarely explored them from a positive perspective except quality of life [11,12,13,14]. In addition, most studies focused on nursing staff rather than all hospital employees. Therefore, it is important to construct an evaluation protocol for the whole person health of all employees in the hospital and provide early referrals and care to promote the whole person health of all medical practitioners.

The purpose of this study is to develop and assess the validity and reliability of “Whole Person Health Scale for Employees of Hospitals (WPHS-EH)” as a measure for whole person health and to subsequently assist hospital administrators in understanding and promoting the health of medical staff.

## 2. Materials and Methods

### 2.1. Study Design and Sample

The researcher explained the study purpose, individual rights, and obtained written consent from all participants. A total of 62 employees were initially engaged in a focus group to construct the WPHS-EH, Subsequently, by applying a cross-sectional study design and convenient sampling, a list of all employees from various departments (about 5000) of MacKay Memorial Hospital in Taiwan was collected, and divided into five strata (doctors, nursing staff, medical technicians, administrative staff, and others) to test the scale. The inclusion criteria were as follows: (1) full-time employees with at least two years of work experience, (2) able to read and communicate in Mandarin, and (3) agreed to participate in the study. During the study period, all participants had no serious physical or mental illnesses. In order to improve the quality participation, five research assistants were responsible for facilitating and guiding the questionnaires from the five departments.

### 2.2. Measures

#### Recruitment Focus Group Participants to Develop and Validate Initial Questionnaire

Demographic characteristics of the research participants include age, gender, department, working years, religion, etc. In preparation for constructing the scale, relevant literature and tool development were summarized and six experts (one doctor, two nurse practitioners, two hospital administrators, and one spiritual caregiver) were invited to discuss and determine the methodologies for the focus group. The interview guidelines include: “1. What do you think whole person health is?”; “2. What factors do you think will promote your health?”; “3. What factors do you think will make you unhealthy?”; and “4. What would you recommend to your colleagues to promote their health?”

From June to August 2014, using the interview guidelines, employees from different departments in the hospital were invited to participate in 2-h focus group discussions. A total of six sessions (62 people) were held, including two sessions among nurses (20 people), two among administrators (17 people), one among doctors (13 people), and one among medical technicians (12 people). The sessions were recorded, transcribed, and subsequently analyzed using ATLAS.ti 7.5 qualitative analysis software that resulted in the identification of 14 questions. Five medical, nursing, and administrative supervisors validated the analysis on two junior high school and one university graduate. It is acceptable for the I-CVI to be at least 0.78, while the S-CVI should be higher than 0.9 [15,16,17], the average item content validity index (I-CVI) and scales content validity index (S-CVI) of questionnaire “adequacy” were 1, while the average I-CVI and S-CVI of questionnaire “clarity” were 0.88 and 1, respectively. The average score for “adequacy” was 3.3 to 3.8, while the average score for “clarity” was 3.0 to 3.5. After expert consultation, items with three points were partially revised for clarity, and the scale was named the “Whole Person Health Scale for Employees of Hospitals” (WPHS-EH). There were 14 questions in the scale, including 9 positive questions (questions 1, 2, 3, 5, 6, 7, 10, 11, and 13) and 5 negative questions (questions 4, 8, 9, 12, and 14). It asks the subjects about their **status of working in the hospital in the past three months**. This scale is scored using a four-point Likert scale, with points 1–4 representing **“Never**,” “**Sometimes**,” “**Often**,” and “**Always**,” respectively, with total scores ranging from 14–56; the higher the score, the better the whole person health of the hospital staff. The questionnaire was revised by experts and conducted by seven employees from different departments in the hospital (one doctor, two nurse practitioners, two administrators, and two medical technicians) for pilot testing, emphasizing the participants’ ability to understand and answer the questions. Reliability and validity tests were subsequently carried out. 

### 2.3. Analysis

SPSS for Windows 22.0 software was used for data analysis and descriptive statistics. The initial scale of each item was first confirmed by the content validity test. Further validation for reliability, which includes Cronbach’s alpha and item analysis, was then performed. A critical ratio (CR) test will be used to select the 27th and 73rd percentile as the cut-off point, representing the low and high groups, respectively. Two independent sample t-tests will then be done to verify the degree of discrimination of each item. Additionally, correlation coefficients between the items and the total score will be conducted. A correlation coefficient less than 0.30 or greater than 0.80 will serve as benchmarks for deleting an item. According to the scale validity test process, exploratory factor analysis (EFA) will include the Kaiser-Meyer-Olkin measure of sampling adequacy, the Bartlett’s test of sphericity, the orthogonal rotation axis, an eigenvalue greater than 1.0, and the scree plot as the standard for extraction factors. In addition, the confirmatory factor analysis (CFA) will be tested for validity using LISREL version 8.83 (Scientific Software International, Inc., Skokie, IL, USA).

## 3. Results

### 3.1. Demographic Characteristics of the Participants

This study was conducted at MacKay Memorial Hospital from June to April 2016. A total of 900 questionnaires were distributed, and 854 were collected (response rate: 94.9%). The questionnaire results of staff from various departments were included in the data analysis. The average participant age was 37.6 ± 10.7 years old. The average work experience was 11.8 ± 9.8 years. The gender distribution was 102 (11.9%) males and 752 (88.1%) females. As for the distribution of staff from various departments, there were 33 doctors (3.9%), 501 nurses (58.7%), 129 medical technicians (15.1%), and 135 administrators (15.8%). (Table 1).

### 3.2. Item Analysis

Item analysis revealed that the average total score was 38.7 ± 5.6 from all employees. In this analysis, items with the highest scores were Q3, “I enjoy the convenience of medical resources”, with a score of 3.2 ± 0.7, and Q1, “I interact well with my colleagues”, with a score of 3.2 ± 0.6. Furthermore, items with the lowest scores were Q4, “I can get meaning from the feedback of the clients served”, with a score of 2.1 ± 0.7, and Q9, “The work-force in my department is insufficient”, with a score of 2.5 ± 0.9. The Skewness and Kurtosis scores were 0.08 and 0.16, respectively, which demonstrate normality. Simultaneously, the item analysis also included the correlation coefficient between the items score and the total score. A correlation coefficient less than 0.30 or greater than 0.80 was deemed the standard for deleting an item. All the values fell between 0.47 and 0.71 in this study, demonstrating a significant correlation. The CR test of each item was discriminant, except Q1 and Q13. The scores between Q1 and Q13 did not show significant differences by independent t-test; however, they did reveal a significant correlation with the total score. Thus, the two items were retained as they pertain to clinical professionalism (Table 2). 

### 3.3. Exploratory Factor Analysis and Reliability

In this study, the Kaiser-Meyer-Olkin measure of sampling adequacy for the 14 questions was 0.89, and the Bartlett’s test of sphericity showed a significant difference (*p* < 0.000), indicating that this scale can be used for factor analysis. Then, the factors extracted by EFA were examined against the following conditions: (1) the factor load of each item should be greater than 0.30 and (2) when each item is extracted by multiple factors, the difference between the two factor loads should be greater than 0.15, the orthogonal rotation axis should be used, the eigenvalue needs to be greater than 1.0, and the scree plot should be used as the standard for the extraction factors [18]. After meeting the above requirements, three factors were extracted via principal component analysis (Table 3). The first component contained six questions (Q2–7) related to the work environment of staff and was named the **“****Hospital circumstance and system.”** The second component contained four questions (Q8, Q9, Q12, and Q14) related to employee workload and risk was named the **“Workload and harm.”** The third component also contained four questions (Q1, Q10, Q11, and Q13) related to personnel relationships, and was named the **“****Professional and interpersonal interaction”** (Table 4). The cumulative explained variation of these three components reached 56.9%.

This study extracted three components through EFA, among which **“****Hospital circumstance and system”** had the highest average score with 16.7 ± 2.5, followed by **“Workload and harm”** with 10.6 ± 2.6, and **“****Professional and interpersonal interaction”** with 11.4 ± 1.9. The internal consistency of these three items with a Cronbach’s alpha value of the **WPHS-EH** was 0.82, while the Cronbach’s alpha values of the three sub-dimensions and total score were 0.78 for **“****Hospital circumstance and system,”** 0.74 for **“Workload and harm,”** and 0.64 for **“****Professional and interpersonal interaction,”** all of which were significantly correlated (Table 4).

### 3.4. Confirmatory Factor Analysis for Construct Validity

In this study, CFA was tested under the structural equation modeling, and the model was evaluated using the unweighted least squares estimate. The index for the overall goodness of fit of the measurement model is as follows: chi-squared/degrees of freedom ratio (χ^2^/df ratio) = *539/105 = 5.1*, GFI = 0.93, AGFI = 0.90, NNFI = 0.95, NFI = 0.95, CFI = 0.96, IFI = 0.96, RFI = 0.94, PNFI = 0.77, and PGFI = 0.66. The results validated the effectiveness of the three-factor model and satisfactory goodness of fit of the overall model of the WPHS-EH (Table 5, Figure 1).

## 4. Discussion

Most studies about employees working in hospitals were focused on nurses, physicians, and other medical professionals, but did not include administrative staff or general staff. Moreover, the focus of research often concerned burnout, demoralization, and fatigue [6,7,8,12]. In Kenya, Muthuri et al. concerned health-related quality of life among healthcare workers to develop and implement national policies and programs for healthcare works [13]. In Taiwan, Fang et al. developed a scale to evaluate the spiritual well-being of physicians, which was considered a medical ethical issue [19,20]. However, the concepts of whole person health involving physical–psychological = social–spiritual dimensions have not received enough attention. The WPHS-EI might be the first tool developed to evaluate whole person health for all healthcare workers, administrative staff, and general staff working in hospitals. The evidence-based tool will develop and implement policies and programs to keep and promote health of employee working in hospitals.

This project was designed to develop and assess the validity and reliability of utilizing the WPHS-EH scale determine overall employee health. In this study, staff from five different departments of a MacKay Memorial Hospital were interviewed by six focus groups. The 14-item **WPHS-EH** scale was determined by expert content validation. Three dimensions were extracted by the EFA: **“****Hospital circumstance and system,” “Workload and harm,”** and **“****Professional and interpersonal interaction.”** The cumulative explained variance of these three dimensions reached 56.9%. The large sample size (*n* = 854) may be reliable to develop the instrument.

This study also evaluated the reliability and validity of the **WPHS-EH**. The results showed that the **WPHS-EH** was a valid and reliable instrument for measuring hospital systems, workload, and relationships between colleagues in this hospital. The internal consistency, inter-subscale correlations, Cronbach’s α, EFA, and CFA of the **WPHS-EH** were investigated. The mean total **WPHS-EH** score was 38.7 (the higher scores was 46) from all employees, which displayed that the Whole Person Health of the staff is very good in this medical center. 

A reliability analysis of internal consistency was conducted. This study found 14 significant item-total correlation coefficients between 0.46–0.71. It is recommended that the average inter-item correlation coefficient should be in the range of 0.15–0.50 [17]. These results indicated that 14 items contributed to the scale. Cronbach’s alpha coefficient was used to test the consistency among items and must be between 0.7–0.9 [16,17]. The WPHS-EH scale was 0.82, while the Cronbach’s alpha values of the three sub-dimensions and total score were 0.78 for “Hospital circumstance and system,” 0.74 for “Workload and harm,” and 0.64 for “Professional and interpersonal interaction.” This indicates that the tools were measuring the same concept, all of which were significantly correlated, indicating that this scale has good reliability.

Regarding of the WPHS-EH model goodness of fit, the results were as follows: (χ^2^/df) =5.1, GFI = 0.93, AGFI = 0.90, NNFI = 0.95, NFI = 0.95, CFI = 0.96, IFI = 0.96, RFI = 0.94, PNFI = 0.77, and PGFI = 0.66. The χ^2^ value is sensitive to the sample size, with larger sample sizes leading to higher χ^2^ values [21]. In this study, the sample sizes were 854, which may have influenced the χ^2^ value. Absolute fit measures, including χ^2^/df, GFI, and AGFI, showed values greater than 0.9. The comparative-fit measures, such as NNFI, NFI, CFI, IFI, and RFI, were greater than 0.9 and were optimal for the model. Additionally, parsimonious-fit measures should ideally have values over 0.5. There results showed suitable results for the increment of fit measures. Overall, the study showed that the WPHS-EH model was an acceptably consistent empirical model, and that the adaptive measurement had overall construct validity. 

Our study has four limitations. First, all participants worked in the same hospital; thus, the study was not a multi-center project. Different hospitals have different hospital cultures and organizations. If WHPS-EH could be tested by other hospitals or other countries, it would be more comprehensive. Second, over half of the participants were nurses. The number of the other occupations were not enough to categorize them for further detailed analysis. Third, there were no other questionnaires to compare the WHPS-EH with. We need further projects to expand the knowledge about whole person heath of employees in hospitals. Last, the WHPS-EH is only a 14-item scale, so it is difficult to cover comprehensive dimensions of the whole person health of employees in hospitals.

Despite these limitations, this research has three implications. First, WHPS-EH was developed from qualitative results from all kinds of employees in the hospital. This means that WHPS-EH can be used to understand the health of employees without discrimination of class or occupation. Second, WHPS-EH has only 14 items, so it can be easily implemented as a screening tool to understand the health conditions of employee in hospitals. Finally, as a self-rating scale, WHPS-EH is a valid tool to arouse the insight of whole person health among those who work in hospitals. Healthcare providers need to remain healthy to ensure that they can provide patients with high-quality medical services.

## 5. Conclusions

This study assessed the WPHS-EH; this scale has good reliability and validity. Its content can evaluate staff opinions in medical institutions in the categories “**Hospital circumstance and system,” “Workload and harm**,” and “**Professional and interpersonal interaction**.” The results can assist in understanding the physical and mental health status of hospital staff and achieve the effect of improving the whole person health status of the staff. This study assessed the WPHS-EH, showing that the scale has good reliability and validity. However, the proportion of nurses is higher in the selected sample, and one medical center may not reflect all hospitals in Taiwan. In addition, although CFA has a good construct validity in this study, future studies can use more golden standard scales to compare and use this scale in different regional hospitals, where its psychometric properties can be tested.

## Figures and Tables

**Figure 1 healthcare-09-00610-f001:**
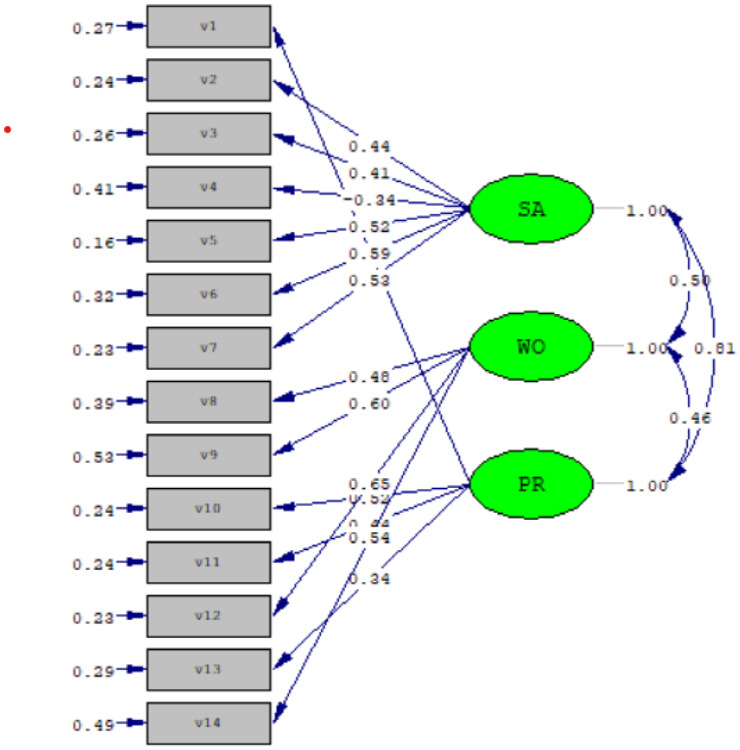
Confirmatory factor analysis of the three-factor model of the WPHS-EH. SA = Hospital circumstance and system, WO = Workload and harm, PR = Professional and interpersonal interaction.

**Table 1 healthcare-09-00610-t001:** Demographic characteristics of the participants (*N* = 854).

Item	*N* (%)	Mean		SD
Age (years)		37.6	±	10.7
Work experience (years)		11.8	±	9.8
Gender				
Male	102 (11.9)			
Female	752 (88.1)			
Working department				
Doctor	33 (3.9)			
Nurse	501 (58.7)			
Medical technician	129 (15.1)			
Administrator	135 (15.8)			
Other	46 (5.4)			
No answer	10 (1.1)			

**Table 2 healthcare-09-00610-t002:** Item analysis of WPHS-EH (*N* = 854).

Questionaire Items	Mean	SD	Skewness		Kurtosis		t	95% Confidence Interval		Correlation
			Statistic	Std. Error	Statistic	Std. Error		Lower	Upper	with Total Score
**Total score of scale**	**38.7**	**5.6**								
Q 1.	3.2	0.6	−0.36	0.084	0.082	0.167	−1.33	−0.16	0.03	0.47 **
Q 2.	3.0	0.7	−0.318	0.084	0.295	0.167	−19.56 **	−1.51	−1.23	0.67 **
Q 3.	3.2	0.7	−0.436	0.084	0.023	0.167	−16.27 **	−1.50	−1.17	0.58 **
Q 4.	2.1	0.7	0.025	0.084	−0.741	0.167	2.05 *	0.01	0.41	0.66 **
Q 5.	2.9	0.7	−0.301	0.084	0.273	0.167	−18.69 **	−1.53	−1.23	0.71 **
Q 6.	2.6	0.8	−0.155	0.084	−0.469	0.167	−24.06 **	−1.96	−1.67	0.67 **
Q 7.	2.8	0.7	−0.120	0.084	−0.324	0.167	−28.34 **	−1.68	−1.46	0.66 **
Q 8.	3.1	0.8	−0.718	0.084	0.346	0.167	−19.59 **	−1.08	−0.88	0.63 **
Q 9.	2.5	0.9	−0.258	0.084	−0.901	0.167	−26.07 **	−1.47	−1.27	0.55 **
Q 10.	2.7	0.7	−0.247	0.084	−0.068	0.167	−9.36 **	−0.59	−0.38	0.69 **
Q 11.	2.7	0.7	−0.141	0.084	−0.096	0.167	−5.32 **	−0.36	−0.17	0.60 **
Q 12.	2.6	0.8	−0.470	0.084	−0.357	0.167	−26.54 **	−1.28	−1.11	0.62 **
Q 13.	2.8	0.6	−0.203	0.084	0.153	0.167	−1.70	−0.18	0.01	0.46 **
Q 14.	2.5	0.9	−0.159	0.084	−0.746	0.167	−25.72 **	−1.37	−1.18	0.55 **

* At a significance level of 0.05 (two-tailed), the correlation was significant. ** At a significance level of 0.01 (two-tailed), the correlation was significant.

**Table 3 healthcare-09-00610-t003:** Rotated factors for principal components analysis of **WPHS-EH**.

Questionaire Items	Factor Loading		
	I	II	III
**Factor 1: Hospital circumstance and system**			
2. My working environment and equipment are suitable	0.558		
3. The medical resources at my hospital are sufficient	0.628		
4. I can get meaning from the feedback from my patients	0.594		
5. I can adapt to the hospital system	0.688		
6. My salary is sufficient	0.771		
7. The welfare provided by the hospital is enough	0.787		
**Factor 2: Workload and harm**			
8. I need to leave my job		0.640	
9. The workforce in my department is insufficient		0.733	
12. My workload is too heavy		0.821	
14. I am at risk of occupational injuries		0.743	
**Factor 3: Professional and interpersonal interaction**			
1. I interact well with my colleagues			0.624
10. I am well balanced in all aspects of my body, mind, and soul			0.592
11. My opinions are heard by others			0.722
13. My professional skillset is utilized			0.736

**Table 4 healthcare-09-00610-t004:** Reliability of the **WPHS-EH** (*N* = 854).

Components	Mean	SD	Cronbach’s α	Correlation between 3 Factors and Total Score ^a^
Total score of the scale	38.7	5.6	0.82	
Factor 1: Hospital circumstance and system	16.7	2.5	0.78	0.82 **
Factor 2: Workload and harm	10.6	2.6	0.74	0.73 **
Factor 3: Professional and interpersonal interaction	11.4	1.9	0.64	0.74 **

^a^ Spearman’s rank correlation coefficient, ** *p* < 0.01.

**Table 5 healthcare-09-00610-t005:** Construct validity of the WPHS-EH.

*Three-Factor Model*	*X*^2^/*df*	*GFI*	*AGFI*	*NNFI*	*NFI*	*CFI*	*IFI*	*RFI*	*PNFI*	*PGFI*
*Scale*	*539/105 = 5.1*	*0.93*	*0.90*	*0.95*	*0.95*	*0.96*	*0.96*	*0.94*	*0.77*	*0.66*

*X*^2^ = Chi-square, *df* = degrees of freedom. *GFI* = Goodness of fit index, *AGFI* = Adjusted goodness of fit index, *NNFI* = Non-normed fit index, *NFI* = Normed fit index, *CFI* = Comparative fit index, *IFI* = Incremental fit index, *RFI* = Relative fit index, *PNFI* = Parsimony normed fit index, *PGFI* = Parsimony goodness of fit index.

## Data Availability

Not report any data.

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
