# Peer review of "Development and Evaluation of Psychometric Properties Regarding the Whole Person Health Scale for Employees of Hospital to Emphasize the Importance of Health Awareness of the Workers in the Hospital"

_healthcare, 2021, doi:10.3390/healthcare9050610_

Round 1

Reviewer 1 Report

The title of the study refers to the overall health of employees. It talks about a scale to assess this health, but I have not detected in the study what level of health the participants had, in order to determine how the factors described influence this level of health. Could you explain this more clearly?

Author Response

Dear Healthcare Editorial Professor:

Please find the original article revised manuscript entitled “Development and Evaluation of Psychometric Properties regarding the Whole-Person Health Scale for Employees of Hospital to Emphasize the Importance of Health Awareness of the Workers in the Hospital” Thank for your time and consideration.

Should we need further efforts to get it published, please feel free to contact us. We will look forward to hearing from you.

Best regard,

Chun-Kai, Fang** and TE-YU, WU ** 

Reviewer 2 Report

The manuscript falls within the scope of the journal. However, it seems to be unclear along some lines making the technical aspects appear weak. Major revisions are recommended.

  1. Title and Introduction

1) The title is ambiguous and needs to be clearly outlined.

2) I believe that the introduction, in particular, is not entirely clear. What is the whole-person health scale? What is the development and evaluation of psychometric properties? It is necessary to answer the questions - what we have studied, for what purpose, what area, and what methods.

3) It is necessary to describe the previous studies and the meaning of the research related to the whole-person health scale.

  1. Materials and Methods

1) The explanation of data variables should be systematically explained in terms of independent variables, dependent variables, and variable types using tables.

2) It is necessary to explain the experimental process using Figure or Table.

3) In Table 1, participants' characteristics need to be described in the form of a crosstab.

4) It is also necessary to explain the reason for the analysis focusing on the reliability and validity analysis.

  1. Results

1) It is not easy to understand because the table's contents and statistical methods are mixed. Please follow the author's guide for how to express the table.

2) It is required to modify the form and contents of Tables 2–4.

  1. Discussion and Conclusion

1) Discussion requires a substantial rewrite to reflect the findings better and reflect on the results in the context of previous studies.

2) Authors should include the usefulness of the whole-person health scale from the results.

3) Authors should also include more limitations and implications of the research.

Author Response

(The authors gave the same response as above.)

Round 2

Reviewer 2 Report

Thank you very much for your revised version of the manuscript. The article is well revised, reflecting the reviewer’s comments. I think the paper meets the standard of the journal, and the revised manuscript is now acceptable.